# The Role of Polygenic Susceptibility on Air Pollution-Associated Asthma between German and Japanese Elderly Women

**DOI:** 10.3390/ijerph19169869

**Published:** 2022-08-10

**Authors:** Sara Kress, Akinori Hara, Claudia Wigmann, Takehiro Sato, Keita Suzuki, Kim-Oanh Pham, Qi Zhao, Ashtyn Areal, Atsushi Tajima, Holger Schwender, Hiroyuki Nakamura, Tamara Schikowski

**Affiliations:** 1IUF—Leibniz Research Institute for Environmental Medicine, Auf’m Hennekamp 50, 40225 Düsseldorf, Germany; 2Medical Research School Düsseldorf, Heinrich Heine University, Universitätsstraße 1, 40225 Düsseldorf, Germany; 3Department of Hygiene and Public Health, Graduate School of Advanced Preventive Medical Sciences, Kanazawa University, 13-1 Takaramachi, Kanazawa 920-8640, Ishikawa, Japan; 4Department of Bioinformatics and Genomics, Graduate School of Advanced Preventive Medical Sciences, Kanazawa University, 13-1 Takaramachi, Kanazawa 920-8640, Ishikawa, Japan; 5Department of Epidemiology, School of Public Health, Cheeloo College of Medicine, Shandong University, 44 West Wenhua Road, Jinan 250012, China; 6Mathematical Institute, Heinrich Heine University, Universitätsstraße 1, 40225 Düsseldorf, Germany

**Keywords:** asthma, air pollution, gene-environment interaction, elderly, ethnicity

## Abstract

Polygenic susceptibility likely influences individual responses to air pollutants and the risk of asthma. We compared the role of polygenic susceptibility on air pollution-associated asthma between German and Japanese women. We investigated women that were enrolled in the German SALIA cohort (n = 771, mean age = 73 years) and the Japanese Shika cohort (n = 847, mean age = 67 years) with known asthma status. Adjusted logistic regression models were used to assess the associations between (1) particulate matter with a median aerodynamic diameter ≤ 2.5μm (PM_2.5_) and nitrogen dioxide (NO_2_), (2) polygenic risk scores (PRS), and (3) gene-environment interactions (G × E) with asthma. We found an increased risk of asthma in Japanese women after exposure to low pollutant levels (PM_2.5_: median = 12.7µg/m^3^, *p*-value < 0.001, NO_2_: median = 8.5µg/m^3^, *p*-value < 0.001) and in German women protective polygenic effects (*p*-value = 0.008). While we found no significant G × E effects, the direction in both groups was that the PRS increased the effect of PM_2.5_ and decreased the effect of NO_2_ on asthma. Our study confirms that exposure to low air pollution levels increases the risk of asthma in Japanese women and indicates polygenic effects in German women; however, there was no evidence of G × E effects. Future genome-wide G × E studies should further explore the role of ethnic-specific polygenic susceptibility to asthma.

## 1. Introduction

The prevalence of asthma has doubled in developed countries over the past 50 years, partly due to environmental exposures [1,2]. Environmental susceptibility to asthma likely differs due to different biological and pathophysiological mechanisms in the life course [2,3,4]. There is a unique age-by-sex interaction effect on asthma diagnosis; there is a higher prevalence of asthma in males before puberty, however after puberty and throughout adult life, there is a higher prevalence of asthma in females [5,6]. Consequently, elderly women are considered a vulnerable subgroup to asthma [5].

The effect of air pollution on asthma can partly be explained by the underlying mechanisms that are related to genetic variations in sex-specific risk loci [2,6]. In addition to sex differences, genetic differences in ethnicities affect susceptibility to diseases such as asthma [7,8,9,10,11,12]. Genetic susceptibilities due to ethnicity are better known in monogenetic diseases such as sickle cell anemia which primarily arose in individuals of African descent, and cystic fibrosis in individuals of European descent [11]. Ethnic-specific susceptibility accompanied by genetic variations such as single nucleotide polymorphisms (SNPs) likely influence individual responses to air pollutants, the risk of asthma, and the efficacy of asthma therapy [7,11,13].

Genetic variants acting along with environmental factors have already been established in gene-environment interaction (G × E) studies [1,2,14]. Existing G × E studies mainly examined childhood-onset asthma [1,2,14,15,16,17], while studies on adults have focused on the candidate genes and SNPs of specific, pathogenic pathways, such as oxidative stress and inflammation [2,14,18]. However, many SNPs, each with a small health effect, are linked within natural synergies across the entire genome. The effects of genome-wide SNPs summarized in a polygenic risk score (PRS) can precisely estimate individual genetic susceptibilities [19,20]. Currently, there are no PRS-air pollution interaction studies of asthma comparing the effects between ethnic groups [2].

Therefore, in this study, we aim to uncover potential differences in the biological mechanisms between ethnic groups by addressing three main questions: (1) How does air pollution exposure affect diagnosed asthma in elderly German and Japanese women? (2) What is the role of polygenic susceptibility? (3) How do G × E effects differ between elderly German and Japanese women? By answering these questions, we aim to further specify and justify asthma prevention strategies according to all ethnic populations [2,3,13].

## 2. Materials and Methods

Data of elderly women from Germany that were enrolled in the “Study on the influence of Air pollution on Lung function, Inflammation, and Aging (SALIA)” and data of elderly women from Japan that were enrolled in the Shika study were used. The characteristics of both studies are presented in Table 1. Asthma was characterized by chronic airway inflammation, reversible airway obstruction, and airway hyper-responsiveness [5,21] and operationalized in both cohorts as a self-report of doctor-diagnosed asthma at a mean age of 73 or 67 years.

Individual exposures to PM_2.5_ and NO_2_ were estimated in both cohorts using average concentrations. In the SALIA cohort, these concentrations were derived from land-use regression models that were assigned within the European Study of Cohorts for Air Pollution Effects (ESCAPE). Further details of the measurements have been described before [26,27] and are summarized in the Online Supplement, p2. In the Shika study, the average concentrations for each participant were estimated using the daily concentrations of each air pollutant that were collected from the Atmospheric Environmental Regional Observation System that were provided by the Japanese Ministry of the Environment and monitoring station data of the Shika area (Online Supplement, p2). To model an exposure window of approximately five years, we used air pollution exposures at the first follow-up examination in the SALIA study, and the mean of averages from baseline and first follow-up examination in the Shika study, statistically centered across the participants. Higher concentrations of air pollution represented higher exposure and were standardized in study sample-specific interquartile ranges (IQR).

Genome-wide genotyping and quality controls [28], e.g., excluding individuals with a different ethnicity than the study sample, were performed and genetic variants were imputed against the Haplotype Reference Consortium reference panel using the Michigan Imputation Server [29] (Online Supplement, p. 2–3). The genetic information was based on the summary statistics from the most appropriate genome-wide association study (GWAS) of asthma in the specific ethnic population [12,20]. Regarding the SALIA cohort, we used a review of GWAS in individuals of Caucasian ethnicity [3], summarizing 128 independent (r^2^ < 0.05) asthma-associated SNPs at a genome-wide significance threshold of *p*-value < 5 × 10^−8^ or *p*-value < 3 × 10^−8^ in more recent studies (Appendix A). Regarding the Shika study, we used the Japanese encyclopedia of genetic associations by Riken (JENGER) database containing summary statistics of approximately 8.7 million SNPs that were tested in the GWAS [30]. From these SNPs, we extracted 11 independent (r^2^ < 0.05) asthma-associated (*p*-value < 3 × 10^−8^) SNPs (Appendix A). Study sample-specific PRS, defined as the individual sum of all risk variant alleles, multiplied by the weight of each allele on the risk of asthma from the most appropriate GWAS, was calculated using the EBPRS R-package [31] (Online Supplement, p. 3). A higher PRS represented a higher number of risk alleles and was standardized in study of sample-specific IQRs.

Descriptive statistics for both study samples, asthma assessments, and air pollution exposures are presented. Logistic regression models were fitted to asthma with a five-year exposure window to PM_2.5_ and NO_2_, respectively, to investigate the main environmental effect while avoiding collinearity between the pollutants. Next, the main polygenic effect was investigated on asthma. Furthermore, the G × E effect was evaluated via a multiplicative interaction term between the air pollutant and the PRS. All the models were adjusted for potential confounders that were selected a priori [4,21] including age, height, weight, education, and ever-/never-smoking (Online Supplement, pp. 3–4). Following the suggestions by Peterson et al. [12], we conducted the G × E analysis in each study sample separately to first investigate the role of polygenic susceptibility on air pollution-associated asthma within the ethnic group and then compared the results across both groups. The results are presented in odds ratios (OR), 95% confidence intervals (CI), and *p*-values. The *p*-values < 0.05 (two-sided) were considered statistically significant. R version 4.1.2 [32] was used for all statistical analyses.

In the sensitivity analyses, descriptive analyses were conducted for participants with genotyped data that were available to reduce the risk of selection bias. Next, we tested the role of binary polygenic risk (high-risk vs. low-risk group), where the group assignment was conducted according to the median of continuous PRS. Furthermore, we tested the G × E effects by applying the genetic risk score-interaction-training approach by Hüls et al. [33] using internal weights from the interaction terms in elastic-net regression models (Online Supplement, p. 3). Additionally, we performed stratified analyses excluding 11 of 771 women that were enrolled in the SALIA study who have changed their residential address in the last five years before the second follow-up examination (at which the asthma status was assessed). The Shika study only included individuals that lived in the Shika town at the time of examination, so residential movers between the examinations were automatically excluded. Finally, we performed stratified G × E analyses to investigate the potential effect modification according to smoking.

## 3. Results

The 771 German women from the SALIA study were on average older than the 847 Japanese women from the Shika study (mean age of 73.5 vs. 67.0 years). Fewer Japanese than German women were diagnosed with asthma (5.9 vs. 8.7%), than ever-smokers (11.5 vs. 19.5%) and had lower individual exposures to PM_2.5_ and NO_2_ five years before the asthma assessments (Table 2). For both ethnic groups, the median air pollution exposures were lower than the annual limits of the European Union (RL 2008/50/EG) for PM_2.5_ (25 µg/m^3^), and NO_2_ (40 µg/m^3^) [34]. However, about the annual air quality guideline levels that were recommended in 2021 by the World Health Organization (WHO) (PM_2.5_: 5 µg/m^3^; NO_2_: 10 µg/m^3^) [35], only the median exposure to NO_2_ in Japanese women was under the limit.

We found harmful environmental effects of PM_2.5_ (OR = 1.2, 95%CI= 0.873;1.695) and NO_2_ (OR = 1.2, 95%CI = 0.927;1.656) on asthma in German women (n = 768), which did not reach statistical significance. However, in the 752 Japanese women, significant adverse effects of PM_2.5_ (OR = 11.7, 95%CI = 3.722;36.909) and NO_2_ (OR = 6.9, 95%CI = 2.441;19.358) on asthma were found.

The significant main polygenic effects were found in German women (n = 531, OR = 0.6, 95%CI = 0.354;0.854), but not in Japanese women (n = 334, OR = 1.5, 95%CI = 0.845;2.532). One IQR increase in PRS decreased the chance of diagnosed asthma by 45% in German women.

No significant G × E effects on asthma were found (Appendix A). The main effects remained stable in the G × E main model. In German women, there were protective polygenic trends when considering the air pollution exposures; in Japanese women, there were adverse effects of both air pollutants when considering the PRS on asthma. In both groups, the direction of the G × E effect was that the PRS increased the effect of PM_2.5_ and decreased the effect of NO_2_ on asthma (Figure 1).

The results of the sensitivity analyses indicate that selection bias should be minimal (Appendix A). The trends were mostly robust for potential effect modifications according to smoking and residential moving (Appendix A). However, the harmful effect of the PRS on PM_2.5_-associated asthma in German women changed to a protective effect in never-smoking women (Figure 1 vs. Appendix A). We found no contradictory G × E results by testing the binary PRS and the genetic risk score-interaction-training approach. However, the polygenic main effects on German women did not reach statistical significance in all the sensitivity analyses (Appendix A).

## 4. Discussion

In this study, we compared the effect of air pollution exposure on diagnosed asthma in German and Japanese elderly women. We further assessed the role of polygenic susceptibility and compared the G × E effects between both ethnic groups. We found an increased risk of asthma in Japanese women after exposure to low PM_2.5_ and NO_2_ levels, respectively, and protective polygenic effects in German women. While we could not find G × E effects on asthma, the main effects remained stable in the G × E model. In both ethnic groups, the direction of the G × E effect was that the PRS increased the effect of PM_2.5_ exposure and decreased the effect of NO_2_ exposure on asthma. The trends were mostly robust to potential effect modifications according to smoking and residential moves.

Asthma is a complex disease with many different forms and comorbidities that are distinguished between sex and age, but not yet between different ethnic groups, such as German and Japanese individuals [21]. When comparing epidemiological indicators, there are higher asthma disability-adjusted life years in Germany than in Japan (201 to 300 vs. 100 to 200 per 100,000 individuals) [5]. This is shown in the elderly women that were enrolled in our study with asthma that were more frequently diagnosed in German individuals when compared to Japanese individuals.

Genetic differences among ethnic groups can affect susceptibility to asthma [7,8,9,10,11,12]. The causes are different frequencies of rare genetic variants between the ethnic groups, where a higher number of rare variants potentially increases the risk of diseases. Historically derived are the different frequencies by the ‘ancestral bottleneck’ which lead to different mixing of ancestries so that individuals of African descent had three times as many rare genetic variants as individuals of Asian and European descent [11]. Analyzing the minor allele frequency (i.e., the frequency of the rare/less frequent genetic variant) of selected SNPs indicated that each ethnic group has its specific risk alleles. Linking these SNPs to the risk of a certain disease uncovers ethnic-specific genetic susceptibilities [8]. Regarding asthma, individuals belonging to different ethnic groups have presented different variants in those genes that encode the receptor that is relevant to asthma drug therapy [11]. In a meta-analysis of case-control studies on associations between asthma and ADAM33, one of the first known asthma candidate genes, different polymorphisms were found among Asian and Caucasian individuals [10]. On a genome-wide level, differences in allele frequencies of asthma-related genes were found between Chinese individuals and other ethnicities such as European individuals [9]. Furthermore, in a multi-ethnic GWAS that included individuals of European and Japanese descent, 673 genome-wide significant asthma SNPs at 16 loci in European populations were found, but no genome-wide significant risk loci were detected in Japanese individuals, possibly due to the lack of statistical power [7].

Consequently, in our study, we used SNPs from GWAS of asthma in the appropriate ethnic population and followed the suggestions by Peterson et al. [12] to conduct the analyses in each ethnic group separately. Our results indicated the polygenic susceptibilities to asthma in elderly German women. The protective effect of the PRS in our study is difficult to compare to other studies that focused on candidate SNPs and different population subgroups; therefore, the protective effect with higher PRS should be further assessed in additional studies. However, considering the high age of the women, one possible explanation for this protective effect could be that most women with a higher PRS have already died and the protective effect of the PRS in the survivors is associated with other protective underlying biological mechanisms. In Japanese individuals in general, polygenic effects could be under- or over-estimated due to scientific bias as genotyping is mostly accurate in people of European descent [12,13,36]. There is a lack of PRS studies in non-European descent participants, as only 19% of all studies included participants of East Asian descent (and only 3.8% of all studies were among other ethnic groups) [37]. Although our study used the Japonica array, specialized for the Japanese population [38], the number of asthma-relevant SNPs from the GWAS that were included in the PRS was limited.

However, not everyone with an increased genetic risk of asthma develops asthma. The increasing prevalence of asthma is associated with environmental exposures [1,2]. A systematic review that analyzed the development of asthma about international immigration confirmed that the environment plays a leading role in asthma development. If individuals from countries with low asthma prevalence, thus individuals with lower ethnic-specific polygenic susceptibility, immigrate into a country with high asthma prevalence, their prevalence at first is lower but changes with longer lengths of stay to a similar prevalence as it is found in natives. [39]. Even air pollution concentrations below the official mean limits have already shown associations with natural-cause mortality [35,40], cause-specific mortality [35], respiratory mortality [35], and respiratory impairment such as asthma [41].

In the Tasmanian Longitudinal Health Study, higher exposures to low NO_2_ concentrations over five years were associated with increased asthma prevalence at ages 45 and 50 years [41]. Our results of elderly Japanese women are consistent with this as we found associations between low exposure to NO_2_ and asthma five years later and added further evidence for PM_2.5_. In contrast, no associations between long-term air pollution and adult-onset asthma were shown in a meta-analysis of five European cohorts besides the SALIA study using standardized ESCAPE exposure estimates [42], which is consistent with our results in elderly German women. In line with Jacquemin et al. [42], one interpretation is that air pollution affects only vulnerable subgroups of adults. Another explanation for the lack of significant environmental effects in elderly German women could be that these women are affected and influenced by higher levels of air pollution at a younger age, as they lived between 1985 and 1994 in the highly industrial Ruhr area, which has decreased the chance of asthma later in older age. It is already known for epigenetic mechanisms that there also exists a critical time-frame early in life for the influence of immunometabolism and allergic diseases [43]. The differences in air pollution effects on asthma between Australian or Japanese and European individuals could indicate underlying ethnic-specific polygenic susceptibilities.

We did not find G × E effects on the pathway of air pollution exposure to asthma in German or Japanese women. The trends in both ethnic groups were the same, as the PRS increased the effect of PM_2.5_ and decreased the effect of NO_2_ on asthma, whereas the harmful trend of the PRS on PM_2.5_ -associated asthma in German women changed to a protective effect in never-smoking women. In a previous review of the G × E effects on adult-onset asthma in European individuals, significant associations between candidate genes and air pollutants (NO_2_ and living less than 200 m away from a major road) were found [2]. Currently, most G × E studies have focused on the associations between childhood-onset asthma [1,2,14,15,16,17] and candidate SNPs that are related to oxidative stress, inflammation, innate immunity, and proteins that are expressed in the bronchial epithelium [2,14,18]. Furthermore, G × E studies of asthma in ethnically Asian populations and among other ethnic groups are lacking [2,12]. To the best of our knowledge, this is the first study on the effect of PRS-air pollution interaction with asthma that compares different ethnic groups, hence an interpretation considering other study results is limited. To expand on this study, future GWAS meta-analysis and genome-wide G × E studies including large multi-ethnic populations are needed to identify the role of ethnic-specific polygenic susceptibility [20,44,45] and provide insights into underlying mechanisms on the pathway of air pollution exposure to asthma. Additionally, epigenetic modifications should be considered in exploring these underlying mechanisms [46]. This evidence will improve precision medicine, disease screening, prevention strategies, and drug development in all ethnic groups [7,11,12,13].

The strengths of our study are the comparability between study characteristics, exposure windows, and genetic processing which reduced the risk of bias across the study samples. We followed recent recommendations to use ethnic-specific genotyping arrays [12,36,38] and to analyze the G × E effects between different ethnic groups using PRS. Sensitivity analyses were performed to confirm our findings, rule out selection bias, and investigate modifying effects of smoking and residential moves.

However, there were also some limitations which should be considered. Despite the comparability between the cohort studies, there are unchangeable differences in the age, examination, and exposure time. Important genetic variants could be missing if asthma-onset was truly in early life, or because not all the identified variants by the GWAS were available in the imputed genetic data [4], or relevant variants only occur in subgroups e.g., individuals in high air pollution areas and thus were unidentified in the GWAS presenting marginal effects [20,44]. In elderly German women, the effect of air pollution and the G × E effect could be underestimated due to the loss of follow-up examination of women with less education, high air pollution exposure, and worse respiratory health. In addition, the lack of statistical power due to the small sample sizes can be a reason for not identifying interaction effects.

## 5. Conclusions

This study found that exposure to low levels of PM_2.5_ and NO_2_ has an adverse effect on asthma in elderly Japanese women. Furthermore, the results indicated a protective polygenic effect on asthma in elderly German women. However, there is no evidence for G × E effects on the pathway of air pollution exposure to asthma in German or Japanese women. Further G × E studies of asthma including genome-wide-derived SNPs are required to explore the role of ethnic-specific polygenic susceptibility. Ethnic groups should be equally involved in genetic studies to uncover potential differences in biological mechanisms to specify and justify the prevention and therapy of asthma.

## Figures and Tables

**Figure 1 ijerph-19-09869-f001:**
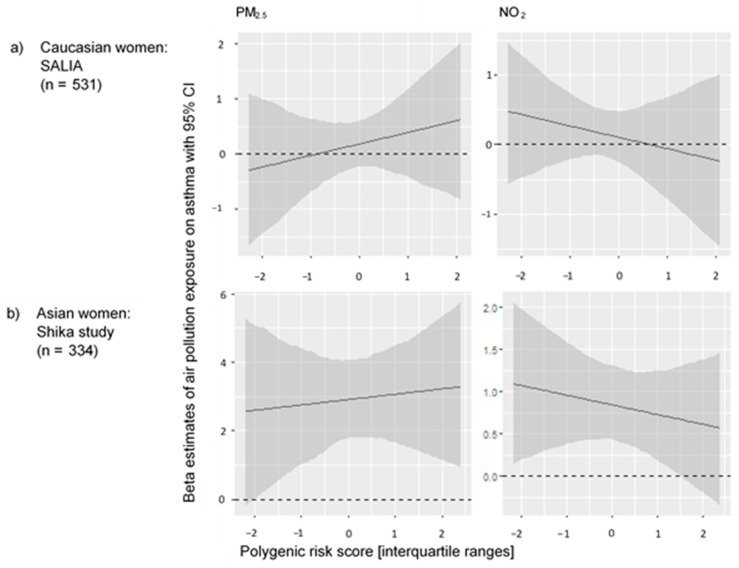
The effects of air pollution exposure on asthma for each sample-specific interquartile range increase of the polygenic risk score in elderly (**a**) German women and (**b**) Japanese women. CI = confidence interval. Adjusted for: age, height, weight, education, and ever-/never-smoking.

**Table 1 ijerph-19-09869-t001:** Cohort study characteristics.

	Cohort Study	Cohort Inclusion Criteria, N	Included Examination in This Study (Year; N)	N, Study Sample Inclusion Criteria (Mean Age)	Ethics Committees and Written Informed Consent from All Participants
**German elderly women**	Study on the influence of Air pollution on Lung function, Inflammation and Aging (SALIA) [22,23]	4874 women aged 55 years living between years 1985 and 1994 in the Ruhr area and the adjacent Münsterland in Germany	1. follow-up (2006; 4027), 2. follow-up (2007–2010; 834)	771 women with information on asthma status at the 2. follow-up (73 years)	Ruhr University, Bochum and the Heinrich Heine University, Düsseldorf
**Japanese elderly women**	Shika study [24,25]	4544 adults aged 40 years or older living between years 2011 and 2016 in the four model areas of the Shika town in Japan	Baseline (2011–2016; 1506), 1. follow-up (2018; 802), 2. follow-up (2019; 245)	847 women with information on asthma status at the 1. and 2. follow-up (67 years)	Kanazawa University, Kanazawa, Ishikawa, Japan

**Table 2 ijerph-19-09869-t002:** Description of the study samples, asthma, and air pollution exposures.

	German Women: SALIA	Japanese Women: Shika Study
**N**	771	847
**Diagnosed asthma (%)**	67 (8.7)	50 (5.9)
**Study characteristics**		
Mean age [years] ± sd	73.5 ± 3.1	67.0 ± 12.9
Mean height [cm] ± sd	163.2 ± 5.8	151.6 ± 6.8
Mean weight [kg] ± sd	72.5 ± 12.4	51.9 ± 8.4
<10 years education (%)	137 (17.8)	393 (46.4)
Ever-smoker (%)	150 (19.5)	97 (11.5)
**Air pollution exposures five years prior to the asthma assessments**		
Median PM_2.5_ exposure [µg/m^3^] (IQR)	17.4 (1.8)	12.7 (3.3)
Median NO_2_ exposure [µg/m^3^] (IQR)	25.9 (9.6)	8.5 (3.6)

Sd = standard deviation, PM_2.5_ = particulate matter with a median aerodynamic diameter ≤ 2.5μm, NO_2_ = nitrogen dioxide, IQR = interquartile range. All women with information on asthma status were included in the analysis.

## Data Availability

The data that are presented in this study are available on request from the corresponding author. The data are not publicly available due to the data protection and privacy laws in the European Union.

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
