# Peer review of "The Role of Polygenic Susceptibility on Air Pollution-Associated Asthma between German and Japanese Elderly Women"

_ijerph, 2022, doi:10.3390/ijerph19169869_

Round 1

Reviewer 1 Report

This manuscript entitled “The role of polygenic susceptibility on air pollution-associated asthma between German and Japanese elderly women” reports a study assessing associating between PM2.5, NO2, PRS, and GxE with asthma in elderly women enrolled in the German SALIA and Japanese Shika cohort. The study appears to be performed, and provides evidence underlining association of adverse effect on asthma in elderly Japanese women with low levels of PM2.5 and NO2 exposure.

My main concern is that the authors utilized 11 asthma-related single nucleotide polymorphisms to calculate the polygenic risk score related to the Japanese women, and draw that no significantly main polygenic effects on Japanese women are observed. For me, this limited number of asthma-related single nucleotide polymorphisms could lead to a bias of significance, although the authors have pointed it out. 

Author Response

Response to Reviewer 1 Comments

This manuscript entitled “The role of polygenic susceptibility on air pollution-associated asthma between German and Japanese elderly women” reports a study assessing associating between PM2.5, NO2, PRS, and GxE with asthma in elderly women enrolled in the German SALIA and Japanese Shika cohort. The study appears to be performed, and provides evidence underlining association of adverse effect on asthma in elderly Japanese women with low levels of PM2.5 and NO2 exposure.

Point 1: My main concern is that the authors utilized 11 asthma-related single nucleotide polymorphisms to calculate the polygenic risk score related to the Japanese women, and draw that no significantly main polygenic effects on Japanese women are observed. For me, this limited number of asthma-related single nucleotide polymorphisms could lead to a bias of significance, although the authors have pointed it out.

Response 1: We thank the reviewer for this useful comment. We understand that the small number of asthma-related SNPs could lead to an underestimate of the polygenic effects as discussed in lines 229-235:

In Japanese individuals in general, polygenic effects could be under- or overestimated due to scientific bias as genotyping is mostly accurate in people of European descent [12,13,36]. There is a lack of PRS studies in non-European descent participants, as only 19 % of all studies included participants of East Asian descent (and only 3.8 % of all studies were among other ethnic groups) [37]. Although our study used the Japonica array, specialized for the Japanese population [38], the number of asthma-relevant SNPs from the GWAS included in the PRS was limited.

Furthermore, we highlighted the limitation as follows (lines 293-297): 

Important genetic variants could be missing if asthma-onset was truly in early life, or because not all identified variants by the GWAS were available in the imputed genetic data [4], or relevant variants only occur in subgroups e.g. individuals in high air pollution areas and thus were unidentified in the GWAS presenting marginal effects [20,44].

To address the concerns of the reviewer we aim to contribute to further studies investigating the non-Caucasian population, including performing large GWAS and genome-wide interaction studies to indicate relevant SNPs for respiratory phenotypes with regard to the environment in specific population groups. This would help to decrease the potential bias in future studies as mentioned in lines 307-310:

Further GxE studies of asthma including genome-wide derived SNPs are required to explore the role of ethnic-specific polygenic susceptibility. Ethnic groups should be equally involved in genetic studies to uncover potential differences in biological mechanisms with the aim to specify and justify the prevention and therapy of asthma.

Reviewer 2 Report

Dear authors, 

I read your manuscript with great interest. Below you will find some minor comments that you may take into account for the final version of your manuscript:

- line 226- concerning the explanation of the protective effect, given the difference of ages between groups,  have you consider to compare groups of German and  Japanese women with similar ages?

- line 256- given the amount of studies that link air pollution exposure to the development of asthma, I don't understand the rationale for the suggestion that  higher levels of air pollution at a younger age may result on a adaptation. 

Best regards

Author Response

Response to Reviewer 2 Comments

Dear authors,

I read your manuscript with great interest. Below you will find some minor comments that you may take into account for the final version of your manuscript:

Point 1: line 226- concerning the explanation of the protective effect, given the difference of ages between groups,  have you consider to compare groups of German and  Japanese women with similar ages?

Response 1: We thank the reviewer for the positive feedback on our manuscript and the useful comments. Due to the study designs of both cohort studies, it is not possible yet to compare the GxE effects in exactly the same age groups as in the SALIA cohort there is information only on the mean ages of 55 and 73 years available. However, we would like to take this suggestion into account after the next follow-up examinations in the Japanese women are finalized as with the higher age of the Japanese women, they are more comparable to the German women age group.

Furthermore, we apologize for any unclear description as we aimed to interpret the protective effect of the PRS in the German elderly women not directly in comparison to the Japanese women. To clarify our explanation of the protective effect of the PRS we adapted lines 226-229 as follows:

However, considering the higher age of the German women, one possible explanation for this protective effect could be that most women with a higher PRS have already died and the protective effect of the PRS in the survivors is associated with other protective underlying biological mechanisms.

Additionally, we highlighted the limitation of age difference as follows (lines 291-293):

Despite the comparability between the cohort studies, there are unchangeable differences in age […].

Point 2: line 256- given the amount of studies that link air pollution exposure to the development of asthma, I don't understand the rationale for the suggestion that higher levels of air pollution at a younger age may result on a adaptation.

Response 2: We thank the reviewer for the careful revision and this comment. We agree that “adaptation” is not the correct explanation and changed lines 255-259 as follows:

Another explanation for the lack of significant environmental effects in elderly German women could be that these women are affected and influenced by atapted to higher levels of air pollution at a younger age, as they lived between 1985 and 1994 in the highly industrial Ruhr area, which has decreased the chance of asthma later in older age.

Reviewer 3 Report

With real interest and pleasure, I read the manuscript ijerph-1814938.

Even though the genetic data are not consistent between Japanese and Germans and no gene by environment interaction has been found, I fully support publication of this work in the International Journal of Environmental Research and Public Health. It is because:

1.       I am a very big fan of genetic studies comparatively performed between populations of different ethnicity. Allele frequencies but mostly LD/haplotypic relationships make it sometimes possible to see/clarify in one population the things not possible to be seen/clarified in the other.

2.       This study is goes much further as it combines genetic predisposition with the environmental exposure, thus fully addressing the asthma development model/background.

3.       An interesting approach is applied with an overall polygenic score based on selected genes/variants but not on single loci-related associations.

4.       The study is generally very well designed and performed.

5.       Furthermore, the data are nicely presented and the paper nicely written.

Some minor/facultative questions/points for the Discussion:

1.       Limitation. The age difference between German (older) and Japanese (younger) women should be mentioned as a limitation, especially as to the best of my knowledge Japanese women are characterized by a higher average life expectancy, right?

2.       Another difference between the cohorts is the time of observation (and exposure?), that seems to be much longer in Germans. It is already partly discussed but could be added to the limitation section if suitable.

3.       Asthma and allergies are prototypic gene x environment diseases, in which the environmental effects modify the genetic factors in a huge part through the epigenetic mechanisms (PMID: 28322581). This should be mentioned.

4.       It is partly discussed but should be highlighted from the environmental/epigenetic point of view. The window of susceptibility/vulnerability and the window of opportunity/intervention are both especially widely open in the childhood, the earlier (including in utero) the wider (PMID: 33668787). Please, discuss how it could affect your results.

Author Response

Response to Reviewer 3 Comments

With real interest and pleasure, I read the manuscript ijerph-1814938.

Even though the genetic data are not consistent between Japanese and Germans and no gene by environment interaction has been found, I fully support publication of this work in the International Journal of Environmental Research and Public Health. It is because:

  1. I am a very big fan of genetic studies comparatively performed between populations of different ethnicity. Allele frequencies but mostly LD/haplotypic relationships make it sometimes possible to see/clarify in one population the things not possible to be seen/clarified in the other.
  2. This study is goes much further as it combines genetic predisposition with the environmental exposure, thus fully addressing the asthma development model/background.
  3. An interesting approach is applied with an overall polygenic score based on selected genes/variants but not on single loci-related associations.
  4. The study is generally very well designed and performed.
  5. Furthermore, the data are nicely presented and the paper nicely written.

We thank the reviewer for this positive feedback. We are delighted to know that our study is of high interest to experts. It highly motivates us to perform further polygenic GxE analysis in different ethnic groups.

Some minor/facultative questions/points for the Discussion:

Point 1: Limitation. The age difference between German (older) and Japanese (younger) women should be mentioned as a limitation, especially as to the best of my knowledge Japanese women are characterized by a higher average life expectancy, right?

Response 1: We thank the reviewer for this useful suggestion and highlighted the limitation of age difference as follows (lines 291-293):

Despite the comparability between the cohort studies, there are unchangeable differences in age, examination, and exposure time.

Point 2: Another difference between the cohorts is the time of observation (and exposure?), that seems to be much longer in Germans. It is already partly discussed but could be added to the limitation section if suitable.

Response 2: We highlighted the limitation of differences in examination and exposure time as follows (lines 291-293):

Despite the comparability between the cohort studies, there are unchangeable differences in age, examination, and exposure time.

Point 3: Asthma and allergies are prototypic gene x environment diseases, in which the environmental effects modify the genetic factors in a huge part through the epigenetic mechanisms (PMID: 28322581). This should be mentioned.

Response 3: We agreed with the reviewer and mentioned the epigenetic mechanisms as suggested in lines follows (lines 277-282):

To expand on this study, future GWAS meta-analysis and genome-wide GxE studies including large multi-ethnic populations are needed to identify the role of ethnic-specific polygenic susceptibility [20,44,45] and provide insights into underlying mechanisms on the pathway of air pollution exposure to asthma. Additionally, epigenetic modifications should be considered in exploring these underlying mechanisms [46].

Point 4: It is partly discussed but should be highlighted from the environmental/epigenetic point of view. The window of susceptibility/vulnerability and the window of opportunity/intervention are both especially widely open in the childhood, the earlier (including in utero) the wider (PMID: 33668787). Please, discuss how it could affect your results.

Response 4: We thank the reviewer for this suggestion and added the epigenetic point of view with its early critical time frame to our discussion as follows (lines 256-261):

Another explanation for the lack of significant environmental effects in elderly German women could be that these women are affected and influenced by adapted to higher levels of air pollution at a younger age, as they lived between 1985 and 1994 in the highly industrial Ruhr area, which has decreased the chance of asthma later in older age. It is already known for epigenetic mechanisms that there also exists a critical time frame early in life for the influence of immunometabolism and allergic diseases [43].